# POUTA - PRODUCE ONCE, UTILIZE TWICE FOR ANOMALY DETECTION

## ABSTRACT

Visual anomaly detection aims at classifying and locating the regions that deviate from the normal appearance. One of the solutions is the reconstruction-based approach, which locates the anomaly by analyzing the difference between the original and reconstructed images. However, when the reconstructed image is of low-quality or the anomaly is fine-grained, the image-level difference analysis approach usually fails. To deal with the above two cases, it is necessary to learn more accurate information. According to our observation, the features of the reconstructive network contains more accurate information about the anomaly than the image-level difference. To leverage the feature-level information, POUTA is proposed. In POUTA, the discriminative network takes the encoder and decoder features of the reconstructive network as the features of the original and reconstructed image respectively. And there is a coarse-to-fine process in each discriminative layer, the above information is refined by the high-level semantics and semantic supervision loss. The discriminative network accepts features as input now, so the feature extraction process (discriminative encoder) is unnecessary. In other words, POUTA produces the features in reconstructive network once but utilizes them twice for reconstruction and discrimination separately, which reduces the parameters and improves the efficiency. The experiments show that, compared with the vanilla method, POUTA achieves better performance with even fewer parameters and less inference time. On MVTec AD, VisA and DAGM dataset, POUTA also outperforms the state-of-the-art reconstruction-based methods.

## 1 INTRODUCTION

Unsupervised anomaly detection has drawn much interest (Yang et al.; Deng & Li, 2022), since it requires no defect samples for training and the collection of defect samples is costly. One of the main approaches of anomaly detection is the reconstruction-based method, which trains a reconstructive network to build a normal version of the input. Generally, the anomaly can be located by analyzing the reconstruction error (Yang et al.; Akcay et al., 2019; Akçay et al., 2019), since there is an obvious difference between the original and reconstructed images in the abnormal region. Recent reconstruction-based methods follow a vanilla network architecture, which contains a reconstructive sub-network and a discriminative sub-network (Yamada et al., 2022; Zhang et al., 2022; Zavrtanik et al., 2021a; Lv et al., 2021). In these methods, the reconstructed image is delivered to the discriminative network which is employed to locate the anomaly by analyzing the difference between the original and reconstructed images.

However, there are two cases in which the image-level difference analysis fails, as shown in Figure 1. Firstly, it is difficult to ensure the reconstructed image to be of high-quality. The reconstructive network sometimes does not repair the abnormal region completely, further leading to detect an incomplete abnormal region, as shown in Figure 1. Secondly, the image-level difference analysis usually fails on the fine-grained anomaly. For example, in the screw of Figure 1, although the abnormal region is completely repaired, the abnormal region shows inconspicuous difference in color before and after reconstruction, and the anomaly escapes the detection of vanilla method. The anomaly of pill in Figure 1 is subtle, which is difficult to locate only using the image-level information. Some previous methods (Wang et al., 2023; Gong et al., 2019; Yang et al., 2020) notice the first issue, and endeavor to build a perfect reconstructed image, so that the image-level difference

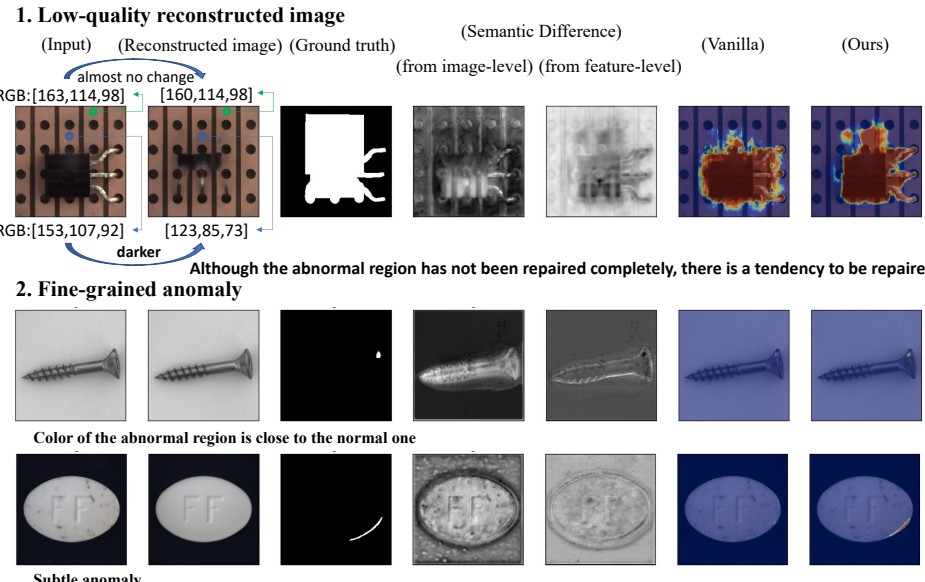

Figure 1: Two cases in which the image-level difference analysis fails. The semantic difference from image-level information is provided by feature map of the discriminative network in the vanilla method. The semantic difference from feature-level is provided by the feature fusion layer of the discriminative network in the proposed POUTA.

is consistent with the abnormal regions. While it is a over-qualified task, since the ultimate goal is to find out the abnormal regions rather than repair the image, and the second issue is still unsolved.

It can be observed that, even for the low-quality reconstructed image, there is a tendency to reconstruct a normal image. As shown in Figure 1, the abnormal reigon of transistor has not been repaired completely, but the corresponding region exhibits a black shadow in the reconstructed image. And as for the normal region, there is no visible difference from the original one. The above phenomenon indicates that although the reconstructive network does not repair the abnormal region completely, it learns the information about where the anomaly is. If this information can be delivered to the discriminative network, it will enables to locate the anomaly more precisely.

Therefore, a natural idea is to utilize the feature-level information in the reconstructive network instead of the image-level information (reconstructed image) to locate the anomaly. Since in the reconstructive network, the features in the encoder are closer to the original input, and the features in the decoder are closer to the reconstructed image, we take the features in the encoder and decoder as the features of the original input and reconstructed image respectively. By analyzing the difference between the symmetric features in the encoder and decoder, the discriminative network is able to obtain the region that tend to be repaired, which reveals the abnormal region. As mentioned above, the discriminative network directly accepts the features from the reconstructive network as the input now, so the feature extraction process is not necessary. Therefore, the encoder of the discriminative network can be removed. That is, our method produce the reconstructive encoder and decoder features once but utilize them twice, which also reduce the parameters and improve the efficiency.

Furthermore, to obtain more refined information about anomaly, in our method, each block of the discriminative network involves a coarse-to-fine process. The current feature will be calibrated by the adjacent high-level feature, and supervised by the semantic supervision loss. Meanwhile, the semantic supervision loss also holds our assumption that the features of encoder and decoder in reconstructive network represent the features of the input and reconstructed image respectively.

Besides, for the fine-grained anomaly, the image-level difference is not significant, but the feature-level semantics in the reconstructive network are obviously different. The semantic difference between the features of the encoder and decoder in the reconstructive network, as well as the semantic difference between the original and reconstructed image are illustrated in Figure 1. It can be seen

that, when only the image-level information is offered, the discriminative network can not obtain the difference between the original and reconstructed image. Therefore the vanilla method fails to detect the fine-grained anomaly. While the feature-level information reveals the fine-grained semantic difference between the original and reconstructed image, further enabling the discriminative network to locate the fine-grained anomaly. The above evidence verifies that the feature-level information in the reconstructive network contains more accurate information about the anomaly than the image-level information.

Our main contributions can be listed as follows:

- We have noticed a phenomenon that, the features in the reconstructive network contains more accurate information about the anomaly than the image-level difference, which enlightens us to propose a new approach for the reconstruction-based anomaly detection.

- A new reconstruction-based method, POUTA, is proposed to directly reuse the symmetric features in the reconstructive network, instead of the conventional image-level difference to locate the anomaly, which improves the performance and also the efficiency.

- POUTA outperforms the state-of-the-art reconstruction-based methods on MVTec AD and VisA dataset. On DAGM dataset, the proposed unsupervised method POUTA achieves a comparable performance with the best-performing supervised method.

## 2 RELATED WORK

The reconstruction-based anomaly methods reconstruct a normal version of the input, usually through the autoencoder (Zavrtanik et al., 2021a;b; Zong et al., 2018; Park et al., 2020). And the anomaly is located by analyzing the differences between the images before and after reconstruction.

Recently, some work (Zavrtanik et al., 2021a; Yamada et al., 2022; Zhang et al., 2022; Zavrtanik et al., 2022; Lv et al., 2021) tends to use the discriminative network to analyze the difference before and after the reconstruction. And the vanilla framework is composed of a reconstructive network and discriminative network in series. For the discriminative network, the information about the normal appearance is only from the reconstructed image provided by the reconstructive network. Therefore, the vanilla method demands the reconstructive network to reconstruct a high-quality normal image, which sometimes is difficult to achieve.

Lots of effort has been expended to improve the quality of the reconstructed image. And most turns to add extra module to the reconstructive network. There are two main purposes for the extra modules. The first is to improve the ability to repair the abnormal region of the input. For the reconstructive network, the abnormal region destroys the normal appearance information in the input. Especially when the anomaly is a large one or a positional anomaly, it is difficult for the reconstructive network to repair the anomaly according to the available information in the input. To address this problem, some use the memory bank (Gong et al., 2019; Park et al., 2020), clustering (Yang et al., 2020) or codebook (Zavrtanik et al., 2022) to record the normal feature and use their combination to replace the input feature, in order to guarantee the decoder accept a normal feature. While these methods cost extra space. Some work (Zavrtanik et al., 2021b; Pirnay & Chai, 2022; Ristea et al., 2022; Madan et al., 2022) take the reconstruction sub-task as an image inpainting task, which grid the image and reconstruct every patch. When a patch is to be reconstructed, it will be removed from the input image. The patch removing aims to reduces the influence of abnormal regions on reconstruction. Recently, it has been proposed to utilize the mathematical expectation map of the training dataset as the supplement of normal appearance (Wang et al., 2023), aiming to assist the reconstructive network to build a high-quality normal image.

And the second is to ensure the normal region of the input to be unchanged. Some work propose to add a decoder (Wang et al., 2023) to reconstruct the input as well, which still remains the anomaly but changes the normal regions to be the same with the reconstructed normal image, which leads to tiny reconstruction error in the normal regions. Also, it has been proposed to solve the problem by limiting the amount of modified pixels during reconstruction (Dehaene et al., 2020), so that the normal regions are more likely to stay unchanged.

The above methods pay attention to the reconstructed image, which follows the vanilla idea that the anomaly is located by analyzing the image-level difference. Therefore, based on the vanilla method,

it will take extra costs to build a high-quality reconstructed image, and still can not deal with the fine-grained anomaly. Our method does not follow the conventional approach, but turns to reuse the features of the reconstructive network, which provides more accurate information about the abnormal region and also improves the efficiency. In this way, our method achieves better performance with less costs than the vanilla method.

## 3 METHOD

As depicted in Figure 2, different with the vanilla method, our method (POUTA) builds the discriminative network directly on the symmetric layers of reconstructive network. The reconstructive network of POUTA follows the design of the vanilla method. And as for the discriminative network, there is no feature extraction process (encoder) since it accepts features of the reconstructive network as input. Each block of the discriminative network involves the coarse-to-fine process. The FCM module treats the features of the encoder and decoder in the reconstructive network as the featrues of the original and reconstruced image respectively. And by analyzing the difference between them, FCM obtains the coarse information about the abnormal region. And HSG refines the information about the abnormal region by introducing the high-level semantic feature to the current feature. Meanwhile, under the supervision of the MSS module, the information about anomaly will be more accurate. Finally, the multi-scale features are concatenated and fused by a feature fusion layer, which is composed by two $3 \times 3$ convolutional layers.

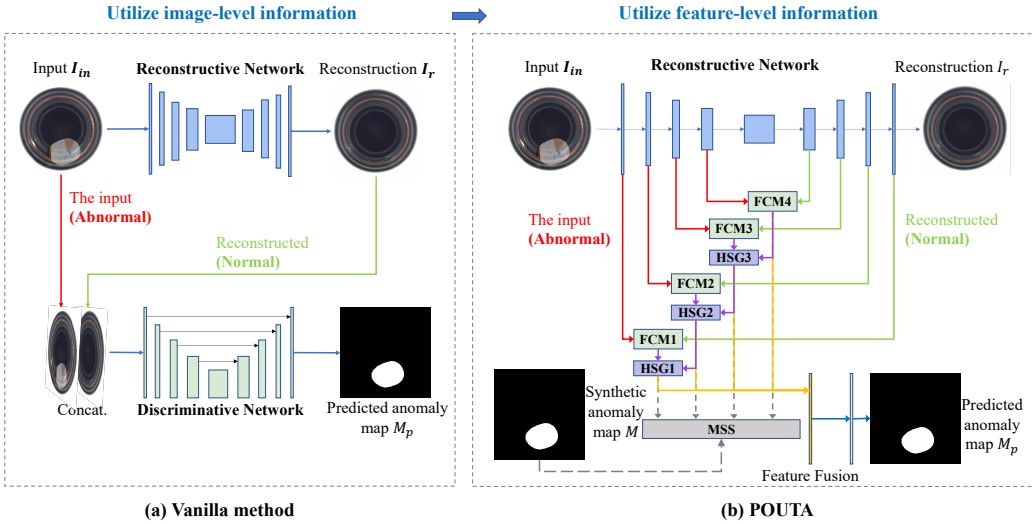

Figure 2: The vanilla method and the proposed POUTA. Vanilla method is composed of an autoencoder-like reconstructive network and a U-Net(Ronneberger et al., 2015)-like discriminative network, where the discriminative network obtains only the images before and after the reconstruction. While in POUTA, the discriminative network obtains the features of the symmetric layers of reconstructive network, and the features contains more accurate information about the abnormal regions than the image-level reconstruction error, which enables the discriminative network to locate the anomaly more accurately. Each block in the discriminative network involves the coarse(FCM module)-to-fine(HSG and MSS module) process.

### 3.1 RECONSTRUCTIVE NETWORK

The reconstructive network is an autoencoder, resembling the vanilla method. There are three parts in reconstructive network: encoder, mapping layer, and decoder. The encoder outputs multi-scale feature $F_{Ei}(i = 1, 2, 3, 4)$ of the input. Then, the mapping layer maps the feature $F_{E4}$ to the normal feature space, and output as $F_M$. Subsequently, the decoder samples the normal feature $F_M$ up, and outputs the multi-scale normal feature $F_{Di}(i = 4, 3, 2, 1)$. Finally, an image with normal appearance $I_r$ is reconstructed.

Since the abnormal images are usually not available during training, a common solution is to generate synthetic anomaly (Zavrtanik et al., 2021a; Liznerski et al., 2020; Zaheer et al., 2020). Similar to (Yang et al., 2023), we firstly use the Perlin generator (Perlin, 1985) and binarization to generate a binary map $M$, which indicates the regions to put synthetic anomaly on. Secondly, an augmented image $I_{aug}$ is obtained by fusing an image from an unrelated dataset (Cimpoi et al., 2014) and a transformed version of the original image (including translation, rotation or identity). Finally, the synthetic abnormal image $I_{in}$ is generated by merging the original image $I_{ori}$ and the augmented image $I_{aug}$ in the regions indicated by $M$. The input image in Figure 2 is an example of the synthetic abnormal image.

Following most reconstruction-based methods(Zavrtanik et al., 2021a; Wang et al., 2023), the reconstructive loss $L_{rec}$ is composed by the mean square error (MSE) and structure similarity index measure (Wang et al., 2004) (SSIM). The reconstructive network is trained to reconstruct the original image $I_{ori}$:

$$L_{rec} = L_{MSE}(I_r, I_{ori}) + L_{SSIM}(I_r, I_{ori}) \tag{1}$$

### 3.2 The Coarse-to-fine discriminative network

The Feature Contrast Module (FCM) generates the coarse information about the anomaly. The High-level Semantic Guidance (HSG) and Multi-scale Semantic Supervision (MSS) refines the information.

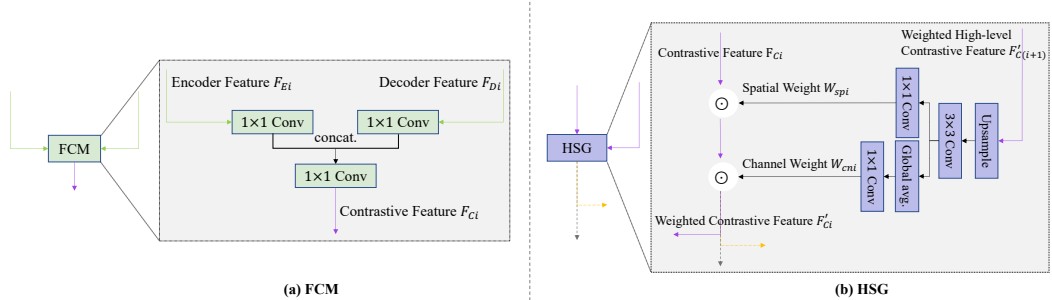

Figure 3: The FCM and HSG modules.

**FCM.** FCM is to analyze the difference between the symmetric features of the reconstructive network. The encoder features $F_{Ei}$ ($i = 1, 2, 3, 4$) and decoder features $F_{Di}$ ($i = 1, 2, 3, 4$) are regarded as the feature of the input and reconstructed image separately. As shown in Figure 3, FCM concatenates $F_{Ei}$ and $F_{Di}$ in channel and outputs the contrastive feature map $F_{Ci}$ ($i = 1, 2, 3, 4$), which contains the coarse information about anomaly.

**HSG.** Generally speaking, the high-level features contains more precise semantic information about the anomaly. To refine the semantic information of each discriminative layer, HSG generates a spacial weight map $W_{sp(i)}$ and a channel-wise weight map $W_{cn(i)}$ from the adjacent high-level feature $F_{C(i+1)}$ (for $i = 3$) or $F'_{C(i+1)}$ (for $i = 1, 2$). And then the two weight maps are broadcasted to the size of $F_{C(i)}$ and multiplied to $F_{C(i)}$, generating a updated contrastive feature $F'_{C(i)}$. And $F'_{C(i)}$ will replace $F_{C(i)}$ to predict the anomaly:

$$W_{sp(i)} = f_{sp(i)}(F'_{C(i+1)}), \quad i = 1, 2, 3 \tag{2}$$

$$W_{cn(i)} = f_{cn(i)}(F'_{C(i+1)}), \quad i = 1, 2, 3 \tag{3}$$

$$F'_{C(i)} = W_{sp(i)} \odot F_{C(i)} \odot W_{cn(i)}, \quad i = 1, 2, 3 \tag{4}$$

where the $f_{sp(i)}$ contains upsample and convolution, and $f_{cn(i)}$ contains upsample, convolution and global average pooling.

**MSS.** Our method assumes the features of the encoder and decoder in the reconstructive network refer to the features of the input and reconstructed image respectively. To hold this assumption and refine the information about anomaly, we propose multi-scale semantic supervision (MSS) module.

MSS requires each weighted contrastive feature map $F'_{C(i)}$ to predict the abnormal regions $M_{p(i)}$ by adding semantic supervision loss to each layer. The semantic supervision loss $L_{SS(i)}$ is composed by L1 loss $L_1$ and focal loss (Lin et al., 2017) $L_{focal}$, since it has been demonstrated that together they can achieve accurate detection performance (Yang et al., 2023). MSS forces our assumption to be held, otherwise $L_{SS(i)}$ will increase and be optimized.

$$L_{SS(i)} = L_{focal}(M_{p(i)}, M) + L_1(M_{p(i)}, M), \ i = 1, 2, 3, 4 \tag{5}$$

where $M$ is a binary map for ground truth, which indicates the abnormal regions. The total loss of MSS $L_{MSS}$ is the weighted sum of $L_{SS(i)}$ on each scale.

$$L_{MSS} = \sum_{i=1}^{4} \lambda_i L_{SS(i)} \tag{6}$$

Finally, the multi-scale features ($F'_{C1}$, $F'_{C2}$, $F'_{C3}$ and $F_{C4}$) are channel-wise concatenated and output a pixel-level predicted anomaly heatmap $M_p$. Consistent with the vanilla method (Zavrtanik et al., 2021a), the image-level anomaly score $S$ is the maximum of the $21 \times 21$ average pooling result of $M_p$. The loss of the predicted heatmap $L_{pre}$ and the total loss of POUTA $L_{total}$ are as follows:

$$L_{pre} = L_{focal}(M_p, M) + L_1(M_p, M) \tag{7}$$

$$L_{total} = L_{rec} + L_{pre} + L_{MSS} \tag{8}$$

## 4 EXPERIMENT

### 4.1 EXPERIMENTAL SETUP

**Dataset.** POUTA is evaluated on MVTec AD (Bergmann et al., 2019), DAGM (Wieler & Hahn, 2007) and VisA (Zou et al., 2022) dataset. MVTec AD is a benchmark dataset for visual anomaly detection, which contains 5,354 images in 15 image categories. The abnormal image is not available in training data, but only in test data. DAGM is a defect detection dataset for supervised methods, which contains 16100 images in 10 categories. There are defect images in training data, while the POUTA is still only trained on the defect-free images and the synthetic anomaly. VisA is a dataset for unsupervised anomaly detection, which contains 10,821 images with 9,621 normal and 1,200 abnormal samples.

**Metric.** Both the image-level classification performance (CLS) and pixel-level location performance (LOC) are evaluated on MVTec AD. The metric to measure CLS and LOC is the area under the receiver operating characteristic curve (AUROC). The average precision (AP) is also used to measure LOC. Since the annotations of DAGM is coarse, only the CLS is evaluated on DAGM. And Consistent with other state-of-the-art method, we evaluate both image-level and pixel-level AUROC on VisA.

**Implementation Details.** $\lambda_1, \lambda_2, \lambda_3$ and $\lambda_4$ in equation 6 is set to 0, 4, 0.3, 0.2 and 0.1 respectively. $\alpha_t$ and $\gamma$ in focal loss is set to 1 and 2 separately. POUTA is trained for 600 epochs with a batch size of 8. The learning rate of the Adam optimizer is set to 0.0002 and multiplied by 0.2 at epoch 480 and epoch 540. Images are resized to $224 \times 224$.

### 4.2 COMPARE WITH THE EXISTING METHODS

The proposed POUTA is compared with several state-of-the-art reconstruction-based methods on MVTec AD, VisA and DAGM, as shown in Table 1 , 2 and 3.

Table 1 shows the quantitative classification (CLS) and location (LOC) results of the state-of-the-art reconstruction-based methods and POUTA on MVTec AD dataset. As for the classification results, POUTA shows higher performance in many cases, and attains a comparable average performance of the SOTA results. And for the average location results, POUTA surpasses the SOTA by 2.9% on AP. In the majority of classes, POUTA achieves the highest or near-highest score, which indicates that POUTA generalizes well over a variety of detection scenarios.

POUTA is further compared with the SOTA reconstruction-based methods on VisA and DAGM dataset. As shown in Table 2, POUTA outperforms the SOTA methods on VisA dataset. And We

| class | DRÆM | | RSTD | | ReContrast | | DeSTSeg | | DBPI | | POUTA | |
|---|---|---|---|---|---|---|---|---|---|---|---|---|
| | CLS | LOC | CLS | LOC | CLS | LOC | CLS | LOC | CLS | LOC | CLS | LOC |
| carpet | 97.0 | 95.5/53.5 | 98.7 | **99.2**/- | **99.8** | -/- | - | 96.1/72.8 | 99.7 | 98.7/**80.6** | 99.1 | 98.0/77.2 |
| grid | 99.9 | **99.7**/65.7 | 100.0 | 99.6/- | 100.0 | -/- | - | 99.1/61.5 | 99.8 | 99.6/69.1 | **100.0** | 99.2/**70.8** |
| leather | 100.0 | 98.6/75.3 | 100.0 | 99.6/- | 100.0 | -/- | - | **99.7**/75.6 | 100.0 | 99.4/**76.0** | 100.0 | 98.4/71.0 |
| tile | 99.6 | 99.2/92.3 | 99.9 | 98.8/- | 99.8 | -/- | - | 98.0/90.0 | 100.0 | 99.5/96.9 | **100.0** | **99.7/97.9** |
| wood | 99.1 | 96.4/77.7 | 99.3 | 98.1/- | 99.0 | -/- | - | 97.7/81.9 | 99.7 | 96.9/78.5 | **100.0** | **98.2/85.4** |
| bottle | 99.2 | 99.1/86.5 | 100.0 | 99.3/- | 100.0 | -/- | - | 99.2/90.3 | 99.9 | 98.9/88.0 | **100.0** | **99.3/91.6** |
| cable | 91.8 | 94.7/52.4 | 99.6 | 98.3/- | **99.8** | -/- | - | 97.3/60.4 | 95.7 | 96.7/66.8 | 97.8 | **98.4/75.9** |
| capsule | 98.5 | 94.3/49.4 | 93.0 | 98.5/- | 97.7 | -/- | - | 99.1/**56.3** | 97.1 | 98.6/48.2 | **99.9** | **99.1**/54.7 |
| hazelnut | 100.0 | **99.7/92.9** | 99.8 | 99.5/- | 100.0 | -/- | - | 99.6/88.4 | 100.0 | 99.5/89.4 | 100.0 | 99.6/88.0 |
| metal nut | 98.7 | **99.5**/96.3 | **100.0** | 98.9/- | **100.0** | -/- | - | 98.6/93.5 | 98.9 | 98.0/92.2 | 99.7 | 99.3/**97.1** |
| pill | **98.9** | 97.6/48.5 | 98.1 | 98.7/- | 98.6 | -/- | - | 98.7/83.1 | 96.6 | **99.4**/88.2 | 98.0 | 99.3/**88.8** |
| screw | 93.9 | 97.6/58.2 | 96.8 | **99.3**/- | 98.0 | -/- | - | 98.5/58.7 | 98.4 | **99.3/60.7** | 98.5 | 97.7/58.1 |
| toothbrush | 100.0 | 98.1/44.7 | 97.9 | 98.3/- | 100.0 | -/- | - | 99.3/75.2 | 100.0 | 99.3/78.1 | **100.0** | **99.5/82.8** |
| transistor | 93.1 | 90.9/50.7 | 98.3 | 90.7/- | 99.7 | -/- | - | 89.1/64.8 | 99.1 | 95.3/72.5 | **100.0** | 97.9/79.5 |
| zipper | 100.0 | 98.8/81.5 | 99.3 | **99.2**/- | 99.5 | -/- | - | 99.1/**85.2** | 98.8 | 98.6/74.1 | **100.0** | 98.9/83.5 |
| avg | 98.0 | 97.3/68.4 | 98.7 | 98.5/- | 99.5 | 98.4/- | - | 97.9/75.8 | 98.9 | 98.5/77.3 | **99.5** | **98.8/80.2** |

Table 1: Image-level classification (CLS) and pixel-level location (LOC) (in %) of the vanilla method DRÆM (Zavrtanik et al., 2021a) and several state-of-the-art reconstruction-based methods (including RSTD (Yamada et al., 2022), ReContrast (Guo et al., 2023), DeSTSeg (Zhang et al., 2022), DBPI (Wang et al., 2023), and our POUTA on MVTec AD dataset (AUROC). The best result for each class is highlighted in bold.

| | PCB1 | PCB2 | PCB3 | PCB4 | Capsules | Candle | Macaroni1 | Macaroni2 | Cashew | Chewing gum | Fryum | Pipe fryum | avg. |
|---|---|---|---|---|---|---|---|---|---|---|---|---|---|
| POUTA | 97.0 | 99.6 | 98.4 | 99.1 | 97.2 | 96.9 | 95.7 | 95.7 | 96.4 | 99.7 | 98.8 | 99.7 | **97.9** |
| EdgRec | - | - | - | - | - | - | - | - | - | - | - | - | 94.2 |
| RD | - | - | - | - | - | - | - | - | - | - | - | - | 96.0 |
| ReContrast | - | - | - | - | - | - | - | - | - | - | - | - | 97.5 |

Table 2: Image-level classification (CLS) (in %) of several state-of-the-art reconstruction-based methods (including EdgRec (Liu et al., 2022), RD(Reverse Distillation) (Deng & Li, 2022), ReContrast (Guo et al., 2023)) and our POUTA on VisA dataset (AUROC). The best result is highlighted in bold.

report the quantitative evaluations on DAGM dataset in Table 3. POUTA outperforms all unsupervised methods and achieves a comparable performance with the best supervised methods. While POUTA requires no real anomaly for training, which enables POUTA more flexible to the situation where the real anomaly is hard to collect.

## 4.3 ABLATION STUDY

To investigate the effect of each component in POUTA, five groups of ablation experiments are conducted on MVTec AD, as shown in Table 4. All the experiments share the same implementation details.

**Image-level information and feature-level information.** To verify the conclusion that the feature-level information from the reconstrctive network is more accurate than the image-level difference, experiments (Vanilla) and (POUTA-base) are conducted in Table 4. (Vanilla) refers to the vanilla method as shown in Figure 2 (a), which utilize only the image-level information to analyze the anomaly. The (POUTA-base) is POUTA without HSG and MSS modules, that is, there is no refined

| | | Class1 | Class2 | Class3 | Class4 | Class5 | Class6 | Class7 | Class8 | Class9 | Class10 | avg. |
|---|---|---|---|---|---|---|---|---|---|---|---|---|
| unsup. | POUTA | 100.0 | 100.0 | 100.0 | 99.9 | 99.9 | 100.0 | 100.0 | 100.0 | 97.4 | 100.0 | **99.7** |
| | DBPI | - | - | - | - | - | - | - | - | - | - | 99.0 |
| | DRÆM | - | - | - | - | - | - | - | - | - | - | 96.0 |
| sup. | CCNN | 99.0 | 100.0 | 98.0 | 99.0 | 100.0 | 100.0 | 100.0 | 100.0 | 100.0 | 100.0 | 99.6 |
| | MS | - | - | - | - | - | - | - | - | - | - | **100.0** |

Table 3: Image-level classification (CLS) (in %) of several state-of-the-art reconstruction-based methods (including DBPI (Wang et al., 2023) and DRÆM (Zavrtanik et al., 2021a)), supervised methods (including CCNN (Racki et al., 2018) and MS (Božič et al., 2021)), and our POUTA on DAGM dataset (AUROC). The best result is highlighted in bold.

| Method | Module | | | average | |
|---|---|---|---|---|---|
| | Fusion | HSG | MSS | CLS | LOC |
| Vanilla | | | | 98.4 | 98.1/75.4 |
| POUTA-base | ✓ | | | 98.8 | 98.1/76.7 |
| POUTA-base+HSG | ✓ | ✓ | | 98.9 | 98.4/76.8 |
| POUTA-base+MSS | ✓ | | ✓ | 98.8 | 98.3/77.2 |
| POUTA | ✓ | ✓ | ✓ | **99.5** | **98.8/80.2** |

Table 4:    Ablation study of POUTA on MVTec AD.

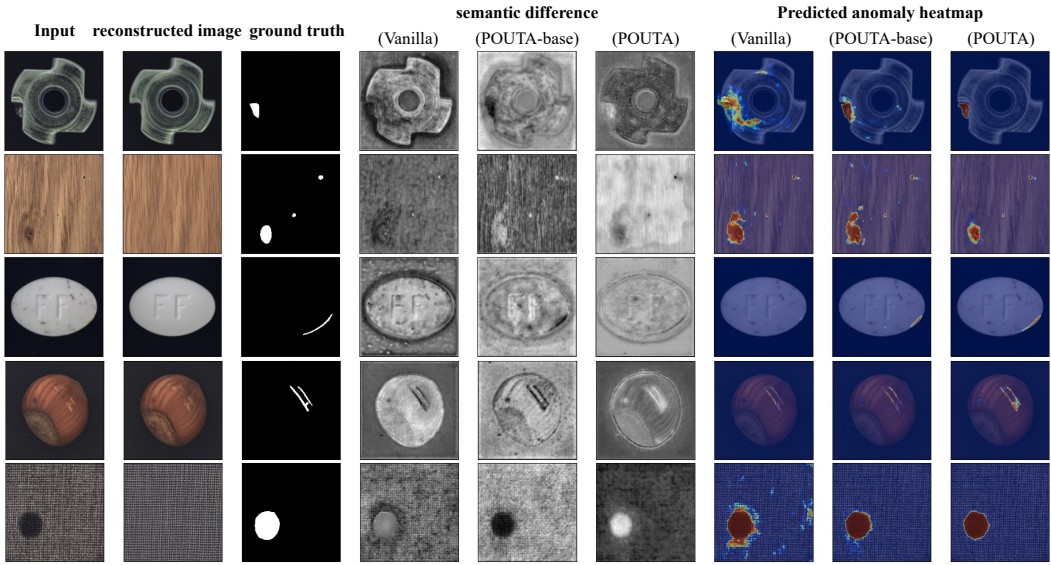

Figure 4: The semantic difference and predicted anomaly heatmap of (Vanilla), (POUTA-base) and (POUTA). The semantic difference of (Vanilla) is provided by the feature in discriminative network of the vanilla method, which is produced by analyzing the image-level difference of the original and reconstructed image. And the feature fusion layer in POUTA provides the semantic difference of (POUTA-base) and (POUTA), which is generated by the feature-level difference of the original and reconstructed image.

step in the discriminative network, the coarse inforamtion from reconstructive network is directly used to locate the anomaly. Table 4 shows that, by utilizing the feature-level information instead of the image-level information, the performance does not suffer from any loss, and even gains a slight improvement, which support our conclusion. Figure 4 demonstrates that the feature-level information provides more fine-grained and accurate semantic difference between the original and reconstructed image, which enables the (POUTA-base) to locate the anomaly more completely and precisely.

**HSG module.** To learn the effect of HSG, experiments (POUTA-base) and (POUTA-base+HSG) are conducted. (POUTA-base+HSG) adds only HSG module to (POUTA-base), and shows little improvement on detection performance as shown in Table 4. Figure 5 demonstrates that the mis-judgment problem are partly relieved after adding HSG. The above results indicate that introducing the high-level semantic information to the low-level feature does benefit to calibrate the semantic information and obtain a more accurate anomaly heatmap. While adding only HSG has limited performance improvement. Due to the lack of MSS, our assumption that the features of the encoder and decoder represent the features of the original and reconstructed image respectively, cannot be guaranteed to hold. The semantic difference between the last few encoder features and the first few decoder features might not be obvious enough, resulting in unclear information about the abnormal region at the high-level semantics, which affects the ability to guide the low-level features.

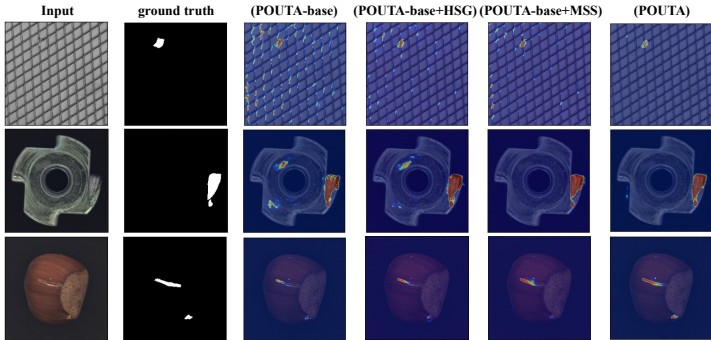

Figure 5: The predicted anomaly heatmap of (POUTA-base), (POUTA-base+HSG), (POUTA-base+MSS) and (POUTA).

|  | POUTA | Vanilla |
|---|---|---|
| Calculation | 35.37G | 39.38G |
| Parameters | 15.02M | 28.68M |
| cost time | 32ms | 52ms |

Table 5: The calculation cost (FLOPs), parameters and cost time of POUTA and vanilla method.

**MSS module.** (POUTA-base+MSS) adds MSS to (POUTA-base). As shown in Table 4, adding MSS improves performance slightly. The misjudgment problem in (POUTA-base) are partially alleviated, as exemplified in Figure 5. MSS requires each layer of discriminative network to predict the anomaly precisely, which does enhance the ability to distinguish the normal and abnormal regions. While without HSG, each layer predicts the anomaly based on the feature of the current scale, which might be not robust and accurate enough since it has not been calibrated by the high-level semantic information.

**The coarse-to-fine process.** (POUTA) adds both HSG and MSS modules to (POUTA-base). There is a significant performance increase as shown in Table 4. And as presented in Figure 4 and 5, the misjudgment problem are significantly mitigated. (POUTA) also shows a noticeable improvement when compared with (POUTA-base+HSG) and (POUTA-base+MSS). With HSG, the semantic information of each layer of the discriminative network is more accurate, since it has been calibrated by the high-level features, further enabling MSS to achieve better supervision effect. And MSS holds our assumption and makes the high-level semantic information about the abnormal regions to be more clear by the segmentation loss. The collaboration between HSG and MSS refines the semantic information about the anomaly, allowing a better performance than adding only one or neither module.

**Parameters, computation and cost time.** We set batchsize to 1, and calculate the average cost time for testing an image by POUTA and vanilla method. Also, the calculation cost (Floating Point Operations Per second, FLOPs) and parameters of each model are calculated, as shown in Table 5. It can be seen that POUTA is more efficient than the vanilla method.

## 5 CONCLUSION

POUTA verifies that the feature-level information in the reconstructive network is more reliable than the image-level difference for reconstruction-based anomaly detection. A new approach, locating the anomaly by analyzing the difference between the encoder and decoder features of the reconstructive network, is proposed, which enables to predict the anomaly more accurately with lower cost than the vanilla method. Moreover, POUTA further refines the feature-level information about the anomaly by HSG and MSS modules in each discriminative layer. Finally, POUTA outperforms the state-of-the-art reconstruction-based methods on MVTec AD, VisA and DAGM dataset, and achieves a comparable results with the best-performing supervised methods on DAGM dataset.

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
