# SUPPLEMENTARY MATERIAL FOR POUTA - PRODUCE ONCE, UTILIZE TWICE FOR ANOMALY DETECTION

We further investigate the effectiveness of the design of HSG in POUTA.

| Method | Module | | | average | |
|---|---|---|---|---|---|
| | Fusion | HSG | MSS | CLS | LOC |
| POUTA-base+implict guidance+MSS | ✓ | | ✓ | 98.8 | 98.5/78.6 |
| POUTA | ✓ | ✓ | ✓ | **99.5** | **98.8/80.2** |

Table 1: The results of the implicit and explicit semantic guidance on MVTec AD.

**Implicit and explicit semantic guidance.** HSG adopts a explict way to guide the lower-level features, which generates a spacial and channel-wise weight map. There is another semantic guidance way, which directly concatenates the previous feature to the lower-level features, guiding the lower-level features implicitly. The explicit way costs more parameters than the implicit one. In order to investigate the distinctions between the implicit and explicit semantic guidance, we conduct experiments (POUTA-base+implict guidance+MSS) and (POUTA) in Table 1. Since the explicit semantic guidance has to generate weight maps, it costs more parameters than the implicit way. But as illustrated in Table 1, the explicit semantic guidance of HSG outperforms the implicit way by a large margin. Figure 1 shows the predicted anomaly heatmap, it can be seen that the implicit semantic guidance is able to improve the accuracy of (POUTA-base) in some cases, while in most cases, the POUTA with the explicit semantic guidance shows more accurate and complete results. The above results demonstrate that it is worth to take more parameters for the explicit semantic guidance.

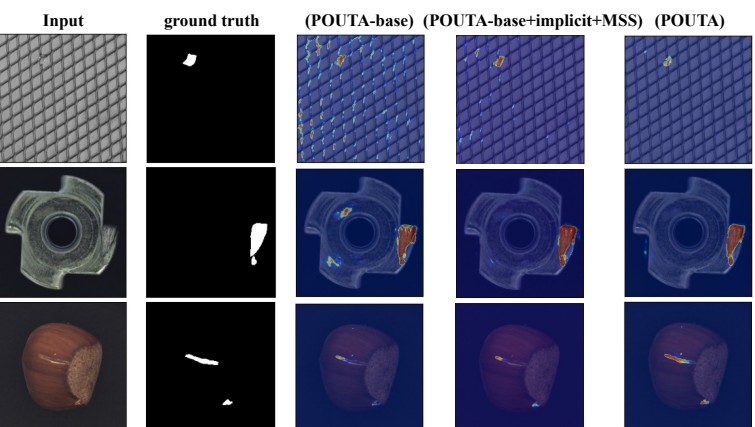

Figure 1: The predicted anomaly heatmap of (POUTA-base), (POUTA-base+implict guidance+MSS), and (POUTA).