# OpenReview forum: "POUTA - Produce once, utilize twice for anomaly detection"
_ICLR.cc/2024/Conference — ICLR 2024 Conference Withdrawn Submission_

### Official Review · Reviewer_9oym · 2023-10-22

**Soundness:** 2 fair
**Presentation:** 1 poor
**Contribution:** 1 poor
**Rating:** 3
**Confidence:** 5

**Summary:**

Targeting dealing with reconstructed image that of low-quality or the anomaly is fine-grained, this paper proposes POUTA, where the discriminative network analysis the features of the reconstructive network, and leverages a coarse-to-fine process. The experiments on MVTec AD, VisA and DAGM dataset show the effectiveness of the proposed method.

**Strengths:**

1.	The architecture is clear, and the motivation sounds reasonable.
2.	Experiments show the effectiveness of the proposed method.

**Weaknesses:**

1.	This paper is lacking important references from the anomaly detection literature. The primary claim made by the authors is that they approach anomaly detection at the feature level rather than the conventional image-level distinction to locate anomalies. However, feature-level reconstruction has already been explored extensively in the literature, including UniAD [a] and its subsequent studies. In these works, the approach of abstaining from reconstructing data at the image level and not using a discriminative network has already been applied. This discrepancy conflicts with the authors' description and literature review.
2.	The ablation studies are carried out on the relatively straightforward MVTec dataset, where the baseline method already achieves an AUC of 98.4%, rendering the comparisons somewhat inconclusive. A more robust evaluation should be conducted on a more challenging dataset like VisA to assess the contributions of each module effectively. Furthermore, the designs of both FCM and HSG appear to be incremental and lack novelty.
3.	This study appears to be outdated, given that there are existing anomaly detection methods that achieve exceptionally high performance on isolated tasks (e.g., 100% AUC on half of the classes in the MVTec dataset). Many recent studies focus on examining how methods perform under a universal model (using a single model for all classes) [a] or in zero-/few-shot settings, such as [b]. It is important to assess how the proposed method performs in these settings as well.
4.	The paper suffers from issues in its writing. It proves challenging to grasp the content, even after multiple readings.

[a] You et al. "A Unified Model for Multi-class Anomaly Detection." NeurIPS 2022.

[b] Jeong et al. "WinCLIP: Zero-/Few-Shot Anomaly Classification and Segmentation." CVPR 2022.

**Questions:**

See the weakness.

---

### Official Review · Reviewer_iu96 · 2023-10-27

**Soundness:** 3 good
**Presentation:** 3 good
**Contribution:** 2 fair
**Rating:** 5
**Confidence:** 4

**Summary:**

This paper studies the undesired failure in anomaly/defect detection. It claims that the image-level difference analysis cannot dealing with the low-quality reconstructed abnormal image and the fine-grained anomaly detection. It is discovered that the features in reconstructive network contains more accurate information about anomaly than the image-level difference. Inspired by this, this paper constructs self-supervised proxy task with synthetic anomaly as segmentation supervision, and designs a serial of subnetworks to adopt feature-level difference analysis. The experimental results outperform existing works or achieve comparable performance, demonstrating the effectiveness of this method.

**Strengths:**

- This paper is well-organized and the motivation is explicit.

- The multi-level feature difference analysis is proved more effective for both detection and localization task. And the evaluation on industry defect detection provides insight into limited discriminant of image-level information.

- Comparing discriminative networks with image-level inputs, the parameters and computational cost are lower without additional feature encoder, a.k.a. produce-once-utilize-twice.

**Weaknesses:**

The major concern with this paper lies in the limited novelty. Despite the feature-level inputs, using discriminative network to identify synthetic anomaly is not new to me. Some similar works exist. For example, the overall framework can be considered as an incremental improvement of DRÆM (Zavrtanik et al., 2021a).

Some claims may not be supported well, e.g., "the discriminative network cannot obtain the difference between the original and reconstructed image with image-level information. And the feature-level information contains more accurate information about the anomaly." Considering the information flow in the reconstruction process, abnormal image contains more anomaly information than the features produced by following encoder, and the normal-like decoded image should also discard more anomaly information. That is, ideally, image-level inputs should be more discriminative.

As mentioned above, I wonder if there is any intuitive explanation as to why DRÆM with more discriminative information cannot be optimized as well as POUTA.

**Questions:**

As mentioned in the "weaknesses" part, I suggest the authors to address my concerns and illustrate the novelty of the proposed work.

---

### Official Review · Reviewer_QRqr · 2023-11-04

**Soundness:** 3 good
**Presentation:** 2 fair
**Contribution:** 2 fair
**Rating:** 5
**Confidence:** 4

**Summary:**

A method for visual anomaly detection and localization is presented. First an auto-encoder network is trained that reconstructs an input. The multi-scale features from the encoders and decoders are then processed by a discriminative network that combines them in a coarse-to-fine fashion and predicts an anomaly map and anomaly score.

**Strengths:**

The work is conceptually similar to previous works in this area like DRAEM. However, instead of comparing pixel-level differences, the authors use features from both the encoding and decoding portions of the reconstructive auto-encoder. This is a simple but sound idea. Results on all the datasets tested are quite good.

**Weaknesses:**

1. The writing is a bit unbalanced: some exposition (especially in Sec. 1, Introduction) can be condensed considerably. On the other hand, some portions lack sufficient technical details. For instance,

    a. What is the architecture of the MSS module? I understand the loss functions used to train it, but there seems to be no description of its architecture.

    b. Further, can the authors please provide additional details about the training regimen? I'm assuming that for each MVTec/DAGM category, a separate set of weights is learnt? Is the whole pipeline (reconstructive network + FCM + HSG + MSS) trained jointly? Or is the reconstructive network trained first, and then the remaining modules (FCM + HSG + MSS) trained from the features of the pre-trained autoencoder?

2. Finally, comparison to non-reconstruction-based approaches is missing. These constitute an important class of methods in the visual anomaly detection domain, and there are several such methods that give state-of-the-art results, a couple of which are listed below:
    - Roth, K., Pemula, L., Zepeda, J., Schölkopf, B., Brox, T. and Gehler, P., 2022. Towards total recall in industrial anomaly detection. In Proceedings of the IEEE/CVF Conference on Computer Vision and Pattern Recognition.
    - Defard, T., Setkov, A., Loesch, A. and Audigier, R., 2021, January. Padim: a patch distribution modeling framework for anomaly detection and localization. In International Conference on Pattern Recognition.

**Questions:**

1. Are the maps $M_{p(i)}$ and $M_p$ heatmaps or binary segmentation maps? Also, I'm assuming that all the $M_{p(i)}$ are resized to the resolution of the ground-truth maps?
2. The losses $L_{pre}$ and $L_{MSS}$ are very similar in formulation; one is applied to features from all levels, while the other is applied to final predicted map. Are both really needed? What is the impact of using only one instead of both?
3. How were the values for various parameters ($\lambda_i, i= 1\dots, 4$; $\gamma$; $\alpha_t$; etc.) in the "Implementation details" chosen? Was it empirically?